A ubiquitination-related risk model for predicting the prognosis and immunotherapy response of gastric adenocarcinoma patients

Shao Shuai 1
Sun Yang 1
Zhao Dongmei 2
Tian Yu 3
Yang Yifan gail1988@163.com yifanyang2023@126.com 4
Luo Nan luonanforeverhom@163.com 5
1 General Surgery, The Second Hospital of Dalian Medical University , Dalian , China
2 Cardiology, The Second Hospital of Dalian Medical University , Dalian , China
3 Vascular Surgery, The Second Hospital of Dalian Medical University , Dalian , China
4 General Surgery, The First Affiliated Hospital of Dalian Medical University , Dalian , China
5 Infection, The Second Hospital of Dalian Medical University , Dalian , China
Zhan Cheng
Electronic publication date: 2024 Jan 31
Publication date: 2024
Volume: 12
Electronic Location ID: e16868
Received 2023 Nov 1; Accepted 2024 Jan 10
Copyright: ©2024 Shao et al.
Copyright year: 2024
Copyright holder: Shao et al.
License: This is an open access article distributed under the terms of the Creative Commons Attribution License, which permits unrestricted use, distribution, reproduction and adaptation in any medium and for any purpose provided that it is properly attributed. For attribution, the original author(s), title, publication source (PeerJ) and either DOI or URL of the article must be cited.
License URL: https://creativecommons.org/licenses/by/4.0/

Keywords: Ubiquitination, Stomach adenocarcinoma, Risk model, Immune infiltration, Immunotherapy

Funding: The authors received no funding for this work.

==============================
Ubiquitination is crucial for the growth of cancer. However, the role of ubiquitination-related genes (URGs) in stomach adenocarcinoma (STAD) remains unclear. Differentially expressed URGs (DE-URGs) were examined in the whole TCGA-STAD dataset, and the prognosis-related genes were discovered from the The Cancer Genome Atlas (TCGA) training set. Prognostic genes were discovered using selection operator regression analysis and absolute least shrinkage (LASSO). A multivariate Cox analysis was further employed, and a polygene-based risk assessment system was established. Signatures were verified using the Gene Expression Omnibus (GEO) database record GSE84433 and the TCGA test set. Using the MEXPRESS dataset, a detailed analysis of gene expression and methylation was carried out. Using the DAVID database, DE-URG function and pathway enrichment was examined. The identified 163 DE-URGs were significantly associated with pathways related to protein ubiquitination, cell cycle, and cancer. A prognostic signature based on 13 DE-URGs was constructed, classifying patients into two risk groups. Compared to low-risk patients, people at high risk had considerably shorter survival times. Cox regression analyses considered prognostic parameters independent of age and risk score and were used to generate nomograms. Calibration curves show good agreement between nomogram predictions and observations. Furthermore, the results of the MEXPRESS analysis indicated that 13 prognostic DE-URGs had an intricate methylation profile. The enhanced Random Forest-based model showed greater efficacy in predicting prognosis, mutation, and immune infiltration. The in vitro validation, including CCK8, EdU, Transwell, and co-culture Transwell, proved that RNF144A was a potent oncogene in STAD and could facilitate the migration of M2 macrophages. In this research, we have created a genetic model based on URGs that can appropriately gauge a patient’s prognosis and immunotherapy response, providing clinicians with a reliable tool for prognostic assessment and supporting clinical treatment decisions.

Introduction

One of the most prevalent gastrointestinal malignancies is stomach adenocarcinoma (STAD), which can be subdivided into cardiac and non-cardiac gastric cancer according to whether the tumor is located near or far from the gastroesophageal junction (cardia) (Colquhoun et al., 2015). Statistically, STAD ranks fifth in global cancer incidence and third in cancer mortality (Ferlay et al., 2015). Individuals with advanced or metastatic sickness have less than 30% of people survive for five years. However, early detection and treatment can raise this to 90–97% (Suzuki et al., 2016). Therefore, more prognostic molecular markers need to be explored.

A role for ubiquitination in mediating protein degradation has been demonstrated. Additionally, ubiquitination contributes to meiosis, autophagy, DNA repair, immunological response, and apoptosis. Most intracellular molecular biological processes involve the ubiquitination proteasome pathway, affecting gene expression and signal transduction to regulate DNA damage repair, participating in aging cell differentiation, regulating tumor malignant transformation, and mediating drug resistance (Popovic, Vucic & Dikic, 2014). Studies have shown that the UbE2D ubiquitin cross-linking enzyme UbE2D33 is associated with human telomerase reverse transcriptase (hTERT), radiosensitivity, and invasiveness in breast cancer. TRIM21 as the E3 ubiquitin ligase TRIM21 may be related to hypopharyngeal tumor cell differentiation (Alomari, 2021). One of its mechanisms is that TRIM21 can up-regulate KRT10 ubiquitination level through non-protein degradation function to improve the stability of KRT10 protein and promote cell differentiation. Two new deubiquitinating enzymatic enzymes, PSMD14 with OTUB1, can inhibit the ubiquitination level of Snail and hinder its proteasomal degradation pathway, thus enabling the accumulation of Snail in cells, and activating the Snail-mediated epithelial-stromal transformation process to promote tumor progression and metastasis in esophageal squamous carcinoma (Jing et al., 2021; Zhou et al., 2018). Therefore, exploring the genes related to ubiquitination in STAD is also necessary. Ubiquitin-specific protease 15 (USP15) regulates the Wnt/β-catenin signaling pathway, aiding in advancing gastric cancer (Zhong et al., 2021). Cells that cause stomach cancer proliferate and spread more quickly when exposed to ubiquitin-associated protein 2 like (UBAP2L) (Lin et al., 2021). The E3 ubiquitin ligase TRIM17 inhibits apoptosis and regulates BAX stability to enhance the survival and progression of gastric cancer (Shen et al., 2023). In addition, OTULIN was recently discovered to be a new biomarker in STAD by developing a ubiquitination-associated signature (Huang et al., 2023). However, a comprehensive model exploring the connection between ubiquitination and prognosis, mutation characteristics, immune activity, and immunotherapy response in STAD has yet to be established.

We created a model based on several prognostic genes and clinical indicators to predict OS in STAD patients. The Cancer Genome Atlas (TCGA) dataset was screened to identify DE-URGs, and the 13 best prognostic gene groups were identified by Cox proportional hazards regression and LASSO univariate hazards regression analyses. Gene expression was multiplied by multivariate Cox coefficients to calculate risk scores. Perform internal/external validation to validate the risk scoring model. Nomograms were created by combining risk scores with clinical parameters and evaluated by calibration plots. DE-URGs were analyzed for their potential biological pathways using functional enrichment analysis. RNF144A from the model was a potent oncogene in STAD and could facilitate the migration of M2 macrophages. Also, RNF144A could determine immunotherapy prediction.

Methods

Data source

RNA-seq data and associated medical details (including survival data) of 32 standard samples and 375 STAD samples were extracted from the TCGA dataset (https://xenabrowser.net). This dataset was used for the preliminary analysis of this study, which is detailed later. In addition, we obtained datasets GSE84433, GSE13911, GSE19826, GSE54129, and GSE65801 according to the GEO database (https://www.ncbi.nlm.nih.gov/). Among them, the GSE84433 dataset containing gene expression profiles of 357 STAD samples with complete survival information was mainly used for external validation of prognostic features, while the GSE13911 (normal: 31; STAD: 38), GSE19826 (normal: 15; STAD: 12), GSE54129 (normal: 21; STAD: 111), and GSE65801 (normal: 32; STAD: 32) datasets were mainly used for expression validation of prognostic genes. In addition, immunohistochemical (IHC) staining images of selected prognostic genes in normal and STAD tissue samples came from the online Human Protein Atlas (HPA; http://www.proteinatlas.org) database.

Furthermore, a total of 669 URGs (Table S1) were obtained from one GO gene set (GO UBIQUITIN DEPENDENT ERAD PATHWAY), one KEGG gene set (KEGG UBIQUITIN MEDIATED PROTEOLYSIS), and three REACTOME genetic sets (UBIQUITINATION OF REACTOME ANTIGEN PROCESSING DEGRADATION OF THE PROTEA, REACTOME DEUBIQUITINATION, and REACTOME PROTEIN UBIQUITINATION) using the MSigDB (Molecular Signatures Database; http://www.gsea-msigdb.org), and only URGs that could be matched in the TCGA-STAD expression profile were utilized in this study.

Identification of DE-URGs

DE-URGs (STAD vs. normal) between TCGA-STAD samples (n = 375) and normal samples (n = 32) were identified based on the R package limma (version 3.5.1). The cutoff was set at P < 0.05 and —log2 fold change (FC)—>0.5.

Functional annotations of DE-URGs

Database-based augmentation and path enrichment analysis for annotating, visualizing, and making artificial discoveries (DAVID, https://david.ncifcrf.gov/) (Jiao et al., 2012). DAVID provides many meaningful annotations for resources from various genomes (Huang da, Sherman & Lempicki, 2009). Biological functions of DE-URG revealed by Gene Ontology (GO) analysis (Harris et al., 2004). The term GO consists of three components: biological process (BP), cellular component (CC), and molecular function (MF). Several biochemical processes involved in DE-URG were inferred using the Kyoto Encyclopedia of Genes and Genomes (KEGG) analysis (Kanehisa et al., 2017). P < 0.05 and count ≥ 2 were considered to be statistically significant.

Selection of prognosis-related genes

Here, we selected 350 samples with complete survival information recorded from the 375 STAD samples in the TCGA dataset for follow-up analysis. The 350 TCGA-STAD samples were randomly assigned 7:3 to a TCGA training set of 245 STAD samples and a TCGA test series of 105 tumor samples. Then, a single-variable Cox regression analysis (P < 0.1) was utilized to find the predictive genes using the TCGA training set’s expression data and the corresponding survival data. LASSO regression analysis also included genes connected to the above prognosis to predict STAD prognosis. To increase the validity and impartiality of the analysis’s findings, we performed 20-fold cross-validation to determine the optimal value of lambda (λ) with the smallest slope probability deviation. The variables obtained from the LASSO regression analysis were identified as the final prognosis-related genes for constructing a multi-gene-based forecasting signature. Univariate Cox and LASSO regression analyses were executed using the R packages Survival (version 3.2-3) and glmnet (version 3.6.1).

Establishment and assessment of the prognostic signature

This research assessed the prognostic validity of a polygenic prognostic signature using a risk-scoring system. The 13 best prognostic genes were identified using multivariate Cox regression analysis, which produced regression coefficients (coef) for each gene. After extracting the expression profiles for the top 13 predicted genes in the TCGA training set, the following procedure was performed to calculate the risk score for each sample. Riskscoresample= ∑n=13ncoefi×xi

where the coefi denotes the regression coefficient of the ith gene, xi denotes the expression value of the ith gene, and n denotes the number of model genes.

Patients in the TCGA training set, depending on the median risk score, are divided into high- and low-risk groups. The variations in survival between the two groups indicated above were examined using K-M survival analysis and a log-rank test. Time-dependent ROC curves can be analyzed using the R package Survival. The multi-gene signature’s prediction ability was also evaluated using ROC.

Furthermore, we pooled samples from the entire TCGA-STAD dataset (n = 350) by risk level to examine the predictive significance of polygenic prognostic signatures influenced by other clinical parameters. K-M curves were used to compare subgroup differences by age, sex, race, tumor stage, pathological TNM stage, and tumor stage.

Validation of the prognostic signature

The predictive potential and generalizability of the multi-gene prognostic signature in STAD were validated using two sets of verification, the TCGA-testing set (n = 105) and the GSE84433 dataset (n = 357). Using the coef of the 13 genes stated above, risk ratings for each patient in the validation set were computed. Based on the dataset’s mean relative risk, the patients were split into high-risk and low-risk categories. Table S2 displays the different metrics of the TCGA-testing set’s STAD specimens. Table S3 shows the pertinent data for the GSE84433 dataset. K-M survival analysis with log-rank test and ROC analysis was performed on the multi-gene prognostic signature.

Independent prognostic analysis

To ascertain whether risk score in STAD patients is a reliable predictive predictor of OS, available variables (age, sex, race, grade, pathological T, N, M, disease stage, risk score), Cox regression analyses were carried out with both single and multiple variables. Variables with P < 0.05 in the Cox univariate analysis were included in the multivariate analysis. A final multivariate variable, P < 0.05, was defined as a unique predictor of outcome for STAD patients.

Construction of the Nomogram

We generated nomograms with independent prognostic factors using the R package root mean square. The survival outcomes (1-, 3-, and 5-year survival rates) anticipated by the Nomogram model were then examined using standard curves to determine whether they were accurately predicted. Survival predicted by the nomogram model and observed outcomes are plotted on the x- and y-axes, respectively, with the 45° line showing the best prediction.

MEXPRESS

DNA methylation profiles of prognostic genes were analyzed using the MEXPRESS platform. Integrate TCGA data with the online tool MEXPRESS (https://mexpress.be/) to visualize methylation profiles based on genomic location and clinical data such as age, weight, and recipient status (Koch et al., 2015; Koch et al., 2019).

In vitro validation

The MKN-7 STAD and THP-1 cell lines were bought from iCell (http://www.icellbioscience.com/search), and cultured in DMEM media and 1,640 media with 10% FBS and 1% double-antibody, respectively. The sequences of RNF144A siRNA are as follows: RNF144A-Homo-1035 GCGCGCAGAUGAUGUGCAATT UUGCACAUCAUCUGCGCGCTT.

RNF144A-Homo-1275 GCAAGUGCAAGUGCAGUAATT UUACUGCACUUGCACUUGCTT.

RNF144A-Homo-667 CCUACAGGAGAACGAGAUUTT AAUCUCGUUCUCCUGUAGGTT. MKN-7 were separated into four groups after siRNA transfection: NC, si-RNF144A-1, si-RNF144A-2, and si-RNF144A-3. After q-PCR (Trizol, 15596026, Thermo Fisher Scientific, Waltham, MA, USA mRNA reverse transcription kit, CWBIO, CW2569; UltraSYBR Mixture, CWBIO, CW2601) validation, the two most potent siRNAs were selected for the subsequent experiments. MKN-7 were separated into three groups for CCK-8 (CCK8 kit, NU679; Dojindo), EdU (EdU kit, RN: R11078.2; RiboBio), Transwell (Transwell chamber, 3428; Corning, Corning, NY, USA), and co-culture Transwell (Transwell chamber, 3428; Corning, Corning, NY, USA) assays after siRNA transfection: NC, si-RNF144A-1, and si-RNF144A-2. The detailed methods are provided in the Supplementary Materials.

Statistical analysis

The R statistical analysis program was used for all calculations. To illustrate the P-values, hazard ratios (HR), and 95% confidence intervals (CI) for each variable, forest plots were produced using the R package Forest Plots. Heatmaps of prognostic gene expression patterns in both high- and low-risk populations were generated using the R package pheatmap. The R package ggplot2 was used to create boxplots showing prognostic genetic makeup production trends in the normal and STAD categories. Wilcox analysis was employed to contrast the prognostic gene expression variations among the normal and STAD subgroups. Test. Pearson correlation coefficient determines the correlation between prognostic genes and inflammatory factors — correlation coefficient—>0.3, critical value P < 0.05. Except as otherwise noted, the cutoff for statistics of importance was P < 0.05.

Results

Analysis of the STAD-related DE-URGs

Based on RNA-Seq data from STAD (n = 375) and normal samples (n = 32) from the TCGA dataset, 617 expression profiles of 669 URGs were extracted (Table S4). Using the R package limma, DE-URGs were found between the STAD and normal groups (STAD vs. normal) based on —log2 FC—>0.5 and Using the R package limma, DE-URGs were found between the STAD and normal groups (STAD vs. normal) based on —log2 FC—>0.5 and P 0.05. As shown in Fig. S1A, 163 DE-URGs were detected; compared to the normal group, 145 were up-regulated, and 18 were down-regulated in the STAD group (Table S5).

Directly afterward, we proposed to perform a study of functional enrichment of the above DE-URGs by DAVID to systematically capture the contribution of these genes in the STAD process. The top 10 terms for each of the three categories in the GO system were presented in Figs. S1B to S1D. Undoubtedly, in the category of BP, DE-URGs were strongly connected with protein ubiquitination-related processes such as protein polyubiquitination, protein deubiquitination, protein autoubiquitination, and the control of protein ubiquitination positively. In addition, these genes were also engaged in embryonic nuclear division, cell proliferation, and other cell cycle-related events. DNA replication, and the G1/S transition of the mitotic cell cycle (negative regulation of mitotic cell cycle progression, positive regulation of ubiquitin-protein ligase activity, and regulation of mitotic cell cycle transition). Furthermore, NIK/NF-B signaling, Wnt signaling (planar cell polarity pathway and the standard Wnt signaling pathway are positively regulated), and other cancer-related phrases were also noticeably elevated. In the CC category, proteasome complex, nucleoplasm, and cytosol were the three most enriched terms, and nucleus (count = 86) was the CC entry with the most DE-URGs involved. The molecular functions of protein binding, ubiquitin-protein ligase activity, ubiquitin-protein transferase activity, ubiquitin-protein ligase activity, and ubiquitin-protein ligase activity were also present in these DE-URGs. The specific outcomes of the GO analysis are displayed in Table S6. KEGG pathway analysis identified a total of 12 pathways that were significantly enriched (Table S7), with ubiquitin mediated proteolysis being the most prominent pathway (Fig. S1E). Moreover, these genes contributed to the cell cycle (oocyte meiosis, progesterone-mediated oocyte maturation, and cell cycle), viral infection (Infection with HTLV-I, viral carcinogenesis, Epstein-Barr virus infection, herpes simplex infection), and cancer (pathways in cancer, small cell lung cancer)-related pathways.

Figure 1 Recognition of prognostic DE-URGs and establishment of the 13 DE-URGs-based prognostic signature in the TCGA-training set.

(A) The DE-URGs’ single-variate Cox regression analysis. (B) Coefficients of prognostic DE-URGs by LASSO regression analysis. (C) Partial likelihood deviance of prognostic DE-URGs by LASSO regression analysis. (D) The association between risk curves and patient survival distributions. (E) The K-M survival curves of the high-risk and low-risk groups. (F) The ROC curves of the risk-scoring system. (G) Heatmap showed the distribution of model genes between the high-risk and low-risk groups.

Recognition of prognostic DE-URGs in the TCGA database

To identify DE-URGs associated with STAD prognosis, we used the TCGA training set’s previously discovered DE-URGs (n = 350) as the basis for a univariate Cox regression analysis. At P < 0.1, we identified 28 out of 163 variables associated with TCGA-STAD survival (Fig. 1A). Subsequently, the LASSO regression analysis with 20-fold cross-validation was used to filter the aforementioned 28 variables, yielding a total of 13 optimum variables, namely BLMH, CUL4A, HCFC1, IDE, NFE2L2, OGT, POLB, PSMD12, RNF144A, TGFBR1, THOP1, TRAF2, and VHL, which were defined as prognostic DE-URGs (Fig. 1B), which were defined as prognostic DE-URGs (Fig. 1C).

Establishment and evaluation of the 13 DE-URGs-based prognostic signatures in the TCGA-training set

We intended to create a new predictive signature for STAD based on the 13 prognostic DE-URGs previously identified. The risk score was calculated using the aforementioned algorithm for each STAD sample (n = 350) in the TCGA training set (Table S8). Figure 1D illustrates risk curves and patient survival distributions. The average risk score of the TCGA training set (n = 122 and n = 123, respectively) was used to divide 350 STAD samples into low-risk and high-risk categories. The K-M survival curve revealed that STAD patients with higher risk levels had a worse prognosis than those with lower risk levels, demonstrating that the risk assessment approach can clearly determine the outlook for STAD sufferers (Fig. 1E). The risk scoring system’s AUCs for projecting STAD patients’ 1-, 3-, and 5-year OS in the TCGA training group were, respectively, 0.665, 0.618, and 0.722, according to the ROC curve (Fig. 1F). This evidence suggested that our 13-gene-based prognostic signature possessed a sure prognostic accuracy. Furthermore, the expression of 13 prognostic genes changed between the groups at high and low risk, as depicted in Fig. 1G. The remaining 10 genes were overexpressed in the high-risk group, while BLMH, RNF144A, and TGFBR1 were comparatively highly expressed in the low-risk group.

Validation of the predictive validity of a 13-gene-based risk scoring system

The predictive validity of the created 13-gene-based risk scoring system in STAD prognosis was validated in both the TCGA testing set and the external validation set (GSE84433 dataset) with the same analytical approach. This was done to ensure the validity and general applicability of the prognostic signature. High-risk STAD patients were consistently associated with poor outcomes in both validation groups (Figs. 2B and 2E). In the TCGA test set (Fig. 2C), the risk scoring system’s AUCs for predicting 1-, 3-, and 5-year OS in STAD patients were 0.648, 0.608, and 0.634; in the external validation, they were 0.601, 0.59, and 0.616 (Fig. 2F). This evidence indicated that the prognostic signature based on the 13 prognostic DE-RRGs possessed tolerable accuracy and applicability for survival prediction for STAD. Furthermore, the risk curves, patient survival distribution, and prognostic gene expression patterns for the two validation cohorts were displayed in Figs. 2A and 2D, respectively. The expression of 13 prognostic genes changed between the high and low-risk groups in the GSE84433 dataset and the TCGA testing set, as depicted in Figs. 2G and 2H.

Figure 2 Verification of the predictive validity of a risk-scoring system based on 13 genes.

(A) The association between risk curves and patient survival distributions in the TCGA-testing set. (B) The TCGA-testing set’s high-risk and low-risk groups’ K-M survival curves. (C) The ROC curves of the risk-scoring system in the TCGA-testing set. (D) The association between risk curves and patient survival distributions in the GSE84433 dataset. (E) The GSE84433 dataset’s K-M survival curves for the high-risk and low-risk categories. (F) The GSE84433 dataset’s ROC curves for the risk-scoring algorithm. (G) In the TCGA-testing set, a heatmap displayed the distribution of model genes between the high-risk and low-risk groups. (H) The GSE84433 dataset’s heatmap displayed the model genes’ distribution between the high-risk and low-risk categories.

Stratified analysis of the 13 gene-based prognostic signature

From the TCGA-STAD database, clinical information, including age, sex, race, tumor grade (grade), pathological T stage, pathological N stage, pathological M stage, and pathological stage (stage), was gathered. Chi-square tests revealed that the risk rating system was highly correlated with race, pathological T stage, and pathological N stage.

We divided STAD patients into clinical tumor stages to determine whether the risk score method’s prediction power extended to additional clinical characteristics, such as age, sex, grade, pathologic stage T, pathological stage N, pathologic stage M, and pathology. K-M analysis showed that the URGs-based signature had significant prognostic value in all subgroups (Fig. 3).

Figure 3 (A–T) Stratified analysis of the 13 gene-based prognostic signature.

The K-M survival curves of the different subgroups in the high-risk and low-risk groups.

An independent prognostic analysis of the 13 DE-URGs-based prognostic signature

According to univariate (Table S9) and multivariate (Table S10) Cox regression analysis, in TCGA-STAD patients, 13 genes and the predictive age signature may be independent prognostic factors for OS (Figs. 4A and 4B). Furthermore, to estimate STAD sufferers’ 1-, 3-, and 5-year OS, we created a Nomogram by merging the two independent prognostic variables mentioned above (Fig. 4C) to provide a more reliable method for predicting clinicians. The presentation of calibration plots suggested that the predicted outcomes of the nomogram pattern for 1- and 3-year OS in STAD patients were in excellent agreement with the actual results; however, we would probably have overestimated its predictive efficiency for 5-year OS in STAD patients (Fig. 4D).

Figure 4 An independent prognostic analysis of the 13 DE-URGs-based prognostic signature.

(A) Univariate Cox regression analysis on clinical characteristics, including the risk-scoring system. (B) Multivariate Cox regression analysis on clinical characteristics, including the risk-scoring system. (C) Nomogram based on clinical characteristics, including the risk-scoring system. (D) 1-year, 3-year, and 5-year OS calibration curves.

Expression validation of the 13 DE-URGs

We extracted the expression profiles of 13 prognostic DE-URGs in the TCGA dataset. Figure S3A displays the expression levels of BLMH, CUL4A, HCFC1, IDE, OGT, POLB, PSMD12, RNF144A, TGFBR1, THOP1, TRAF2, and VHL were relatively high in the STAD group (n = 375); while NFE2L2 was overexpressed in the normal group (n = 32). We then also extracted the expression profiles of these 13 genes in datasets GSE13911, GSE19826, GSE54129, and GSE65801. The different expression trends of BLMH, CUL4A, NFE2L2, POLB, PSMD12, TGFBR1, and THOP1 in the GSE13911 dataset between the STAD group (n = 38) and the normal group (n = 31) were in line with the outcomes of the TCGA dataset (Fig. S3B); The differential expression trends of BLMH, NFE2L2, RNF144A, and THOP1 between the STAD (n = 12) and normal (n = 15) groups in the GSE19826 dataset were consistent with TCGA dataset results (Fig. S3C); BLMH, HCFC1, IDE, NFE2L2, POLB, RNF144A, TGFBR1, and TRAF2 in the GSE54129 dataset showed a consistent differential expression trend between the STAD (n = 111) and normal (n = 21) groups with TCGA dataset results (Fig. S3D); In the GSE65801 data set, the trend of differential expression of BLMH, CUL4A, HCFC1, POLB, PSMD12, RNF144A, THOP1, and TRAF2 in STAD group (n = 32) and normal group (n = 32) is consistent with the results of TCGA (Fig. S3E). The Human Protein Atlas database also validated protein expression for 13 genes (Fig. S2).

Methylation analysis of the 13 prognostic DE-URGs

Research has shown that the CpG island area of the gene promoter region’s DNA methylation corresponds with gene expression (Katopodis et al., 2021; Milella et al., 2015). Therefore, 13 prognostic genes’ methylation status was examined using MEXPRESS in 32 healthy and 375 tumor tissues. Briefly, a total of eight CpG sites were found to be significantly associated with BLMH expression (Pearson correlation coefficients from −0.444 to 0.252. Table S11); 28 CpG sites were significantly associated with the expression of CUL4A (Pearson correlation coefficients from −0.443 to 0.446; Table S12); five CpG sites were significantly associated with the expression of HCFC1 (Pearson correlation coefficients from −0.108 to 0.167; Table S13); 13 CpG sites were significantly associated with IDE expression (Pearson correlation coefficients from −0.359 to 0.324; Table S14); 12 CpG sites were significantly associated with the expression of NFE2L2 (Pearson correlation coefficients from −0.231 to 0.125; Table S15); four CpG sites were significantly associated with the expression of OGT (Pearson correlation coefficients from −0.204 to 0.257; Table S16); six CpG sites were significantly correlated with POLB expression (Pearson correlation coefficients from −0.194 to 0.215; Table S17); eight CpG sites were significantly associated with PSMD12 expression (Pearson correlation coefficients from −0.249 to 0.113; Table S18); 25 CpG sites were significantly associated with the expression of RNF144A (Pearson correlation coefficient from −0.427 to 0.448; Table S19); six CpG sites were significantly associated with TGFBR1 expression (Pearson correlation coefficients from −0.271 to 0.216; Table S20); a 20 CpG sites were significantly associated with THOP expression (Pearson correlation coefficients from −0.295 to 0.339; Table S21). 18 CpG sites were significantly associated with TRAF2 expression (Pearson correlation coefficients from −0.339 to 0.257; Table S22), and seven CpG sites were significantly correlated with VHL expression (Pearson correlation coefficients from −0.215 to 0.266; Table S23).

Correlation analysis of prognostic genes and inflammatory factors

One of the ubiquitin system’s primary roles is controlling inflammation (Lopez-Castejon, 2020). The classical activation pathway of NF-κB signaling has been identified to regulate the progression of gastrointestinal malignancies associated with inflammation (Merga et al., 2016). We obtained nine inflammatory factors of the NF- κB signaling pathway from the NCBI database, namely IL1B, IL6, TNF, NOS2, ISYNA1, IL17A, TGFB1, IL23A, and NLRP3. Pearson correlation analysis (Table S24) showed that RNF144A was positively associated with three inflammatory factors (TGFB1, NLRP3, and ISYNA1) (correlation coefficient between 0.339 and 0.442, all P < 0.05); POLB is positively correlated with IL23A (correlation coefficient = 0.300, P = 6.25E-10). Besides, box plots showed that IL23A and ISYNA1 expression levels were markedly increased in STAD (P < 0.0001; Fig. S3F).

An enhanced prognostic signature based on the Random Forest algorithm

To increase the reliability and accuracy of our risk-scoring system, a random forest algorithm was applied to the 13 prognostic DE-URGs (Zang et al., 2022). The distribution of risk-scoring systems in different datasets is shown in Fig. 5A. The Cox regression and C-index of the risk-scoring system in different datasets revealed the high efficiency of the risk-scoring system in the TCGA dataset (Fig. 5B). High-risk STAD patients were associated with poor outcomes in the TCGA dataset (Fig. 5C). The risk scoring system’s AUCs for predicting 1-, 3-, and 5-year OS in STAD patients were 0.972, 0.978, and 0.966 in the TCGA dataset (Fig. 5D). The calibration curves of the risk-scoring system in the TCGA dataset are shown in Fig. 5E. The decision curves of the risk-scoring system in the TCGA dataset are shown in Fig. 5F. CNV analysis of the risk-scoring system in the TCGA dataset revealed that oncogenic genes (PCLO, 18q11.2, 4q22.1) were relatively highly mutated in high-risk STAD patients (Fig. 5G). Function annotation of the enhanced prognostic signature is conducted in Figs. S4 to S5. GO enrichment analysis revealed that the cellular metabolic process, intracellular anatomical structure, cytoplasm, and intracellular membrane-bounded organelles were highly enriched. KEGG enrichment analysis revealed that metabolic pathways were highly enriched.

Figure 5 An enhanced prognostic signature based on the Random Forest algorithm.

(A) The distribution of risk-scoring systems in different datasets. (B) The Cox regression and C-index of the risk-scoring system in different datasets. (C) The TCGA dataset’s K-M survival curves for the high-risk and low-risk categories. (D) The ROC curves of the risk-scoring system in the TCGA dataset. (E) The calibration curves of the risk-scoring system in the TCGA dataset. (F) The decision curves of the risk-scoring system in the TCGA dataset. (G) CNV analysis of the risk-scoring system in the TCGA dataset.

The function annotation and drug prediction of RNF144A

Among the 13 prognostic DE-URGs, RNF144A was regarded as the most potent prognostic gene that STAD has not thoroughly explored. RNF144A was associated with the immune infiltration of DCs, T cells, fibroblasts, endothelial cells, and macrophages (Fig. 6A). RNF144A was associated with immune modulators such as CD276, HAVCR2, TGFB1, CD86, and CCL2. Drug prediction from GDSC1, GDSC2, CTRP, and PRISM related to RNF144A showed that the drug sensitivity of perhexiline, TAF1_5496_1732, trifluoperazine, PF-00299804_363 were associated with RNF144A (Figs. 6C–6F).

Figure 6 Function annotation of RNF144A.

(A) Immune infiltration related to RNF144A. (B) Immune modulator related to RNF144A. (C) Drug prediction from GDSC1 related to RNF144A. (D) Drug prediction from GDSC2 related to RNF144A. (E) Drug prediction from CTRP related to RNF144A. (F) Drug prediction from PRISM related to RNF144A.

Immunotherapy prediction of RNF144A

Given that RNF144A was closely connected to immune infiltration, the immunotherapy determinant role of RNF144A was speculated. The ROC curves of RNF144A in different immunotherapy datasets were 0.643 (Ascierto cohort), 0.769 (Lauss cohort), 0.733 (Homet cohort), 0.818 (Cho cohort), 0.849 (Kim cohort), and 0.625 (VanAllen cohort) (Fig. 7A). High-risk patients were associated with better outcomes in the Hugo and VanAllen cohorts (Fig. 7B). High-risk STAD patients were associated with poor outcomes in the Lauss, Kim, Nathanson, and Cho cohorts (Fig. 7B).

Figure 7 Immunotherapy prediction of RNF144A.

(A) The ROC curves of RNF144A in different immunotherapy datasets. (B) The immunotherapy datasets’ K-M survival curves for the high-RNF144A and low-RNF144A categories.

Experimental validation on RNF144A

Tests conducted in vitro to demonstrate RNF144A’s likely biological activity were spurred by its unexpected role in STAD’s tumorigenic and immunogenic processes. Figure 8A illustrates how three si-RNF144A groups’ RNF144A expression in the MKN-7 cell line was significantly lower than that of the normal control (NC) group. Subsequent research was done with si-RNF144A-1 and si-RNF144A-2, which showed a markedly improved capacity to obstruct RNF144A expression. As shown in Fig. 8B, in MKN-7, the OD values obtained from the CCK-8 test were significantly decreased in the si-RNF144A-1 and si-RNF144A-2 groups compared to the NC group. Furthermore, compared to the NC group, the si-RNF144A-1 and si-RNF144A-2 groups showed a significant decrease in the quantity of positively stained cells seen in the EdU experiment, as shown in Figs. 8C and 8D. These findings suggested that RNF144A might have an impact on STAD growth. In MKN-7, Transwell results revealed that the migratory STAD cells were considerably suppressed in the si-RNF144A-1 and si-RNF144A-2 groups compared to the NC group (Figs. 8E and 8F), suggesting that RNF144A could affect STAD migration. As demonstrated in Figs. 8G and 8H, the co-culture Transwell test in the MKN-7 trial showed a significant decrease in migratory M2 macrophages in both the si-RNF144A-1 and si-RNF144A-2 groups compared to the NC group. According to this, RNF144A may affect M2 macrophage migration in STAD.

Figure 8 The promoting role of RNF144A in tumor proliferation.

(A) q-PCR results of the relative RNA expression of RNF144A in four groups (NC, si-RNF144A-1, si-RNF144A-2, si-RNF144A-3) in MKN-7. Statistical results were based on student t-tests. (B) CCK-8 results of the OD values in three groups (NC, si-RNF144A-1, si-RNF144A-2) in MKN-7. Statistical results were based on two-way analysis of variance (ANOVA). (C) EdU results of the positively stained cells in three groups (NC, si-RNF144A-1, si-RNF144A-2) in MKN-7. (D) Statistical results of EdU were based on student t-tests. (E) Transwell results of the migrated cells in three groups (NC, si-RNF144A-1, si-RNF144A-2) in MKN-7. (F) Statistical results of Transwell were based on student t-tests. (G) Co-culture Transwell results of the migrated cells in three groups (NC, si-RNF144A-1, si-RNF144A-2) in MKN-7. (H) Statistical results of co-culture Transwell were based on student t-tests.

Discussion

A highly conserved protein is ubiquitin, which is present in all eukaryotic cells. Ubiquitin regulates ubiquitination by covalently binding to target proteins reversibly. Additionally, to being engaged in biological processes like signal transmission, cell cycle regulation, and gene transcription and translation, ubiquitination is essential for destroying proteins. The ubiquitin activator cascade mediates ubiquitinase E1-ubiquitin conjugating enzyme E2-ubiquitin ligase E3 (Buetow & Huang, 2016; Schulman & Harper, 2009; Ye & Rape, 2009). In a highly diverse community, one of the biggest causes of cancer-related death is stomach cancer. It is essential to develop specific prognostic models to improve treatment strategies. Abnormal activation of the ubiquitin-proteasome system in gastric cancer is connected to the emergence and growth of gastric cancer.

There are several prognostic models for STAD. Each model has its advantages and disadvantages. The prediction model based on multiple biomarkers is more effective and accurate. This study was based on screening differentially expressed URGs in the entire TCGA-STAD dataset. Finally, the 13 optimal variables were determined. Specifically, BLMH, CUL4A, HCFC1, IDE, OGT, POLB, PSMD12, RNF144A, TGFBR1, THOP1, TRAF2, and VHL, and the COX risk prediction model was established accordingly. CUL4A, as the main skeleton of the E3 ligase core structure, can bind to DDB1 (Damage-specific DNA binding protein 1) to ubiquitinate various substrates and mediate their degradation, thus affecting protein levels in cells (Zhang et al., 2021). Numerous malignancies exhibit high levels of CUL4A expression, according to studies. CUL4A plays critical roles in cell growth, proliferation, differentiation, senescence, and metabolism through UPS-mediated substrate degradation (Bai et al., 2022). It was found that CUL4A and nuclear transcription factor NF-κB were overexpressed in both gastric and pancreatic cancer tissues, and the expression of CUL4A and atomic transcription factor NF-κB was positively correlated in tumor tissues, suggesting that CUL4A may promote the invasion of malignant tumors by regulating NF- κB signaling pathway (Gong et al., 2017). Therefore, it can be speculated that CUL4A promotes the encroachment and spread of malignant cells through this pathway in gastric cancer.

TGFBR1 is targeted for degradation or cleavage by ubiquitin, thereby lowering TGF-signaling or promoting target gene expression in the nucleus (Zhang et al., 2012). TGFBR1 is ubiquitinated and degraded by the E3 ligases SMURF1, SMURF2, NEDD4-2, or WWP1 when SMAD7 binds with active TGFBR1, lowering downstream signaling.

Interestingly, TRAF6 also encourages TNF-converting enzyme (TACE) and presenilin-1 to break down polyubiquitinated TGFBR1 by proteolysis, freeing the intracellular domain for nuclear translocation (Mu et al., 2011). The intracellular domain, including Snail and MMP2, activates a gene group that encourages cancer cell invasion. TRAF6 is essential for non-canonical TGF-β signaling. TRAF4 modulates canonical signaling in breast cancer cells by ubiquitinating SMURF2 for destruction and mediates non-canonical signaling in a Traf6-independent manner, stabilizing TGFBR at the plasma membrane.

The ubiquitin-proteasome system is an intracellular pathway that relies on ATP rather than lysosome to hydrolyze proteins and selectively degrades cycle-regulating proteins, oncogenes, and tumor suppressor genes. There are also deubiquitination proteases that regulate protein ubiquitination and affect protein hydrolysis. The occurrence of gastric cancer requires the combined action of oncogene activation and related factors promoting tumor cell growth. A critical function of the ubiquitin-proteasome system is to promote healthy cell growth and renewal. On the one hand, it rapidly degrades regulatory proteins. On the other hand, it selectively degrades proteins that cannot fold correctly due to mutation or are damaged after synthesis. It is crucial for controlling protein homeostasis in vivo. The involvement of the ubiquitin-proteasome system in antigen processing, stress response, protein modification, and other factors influence the development and occurrence of disorders of the neurological system, skin diseases, cystic fibrosis, atherosclerosis, rheumatism, self-disease-free diseases, and tumors.

The Wnt/β-catenin and hedgehog signaling pathways have been involved in regulating tissue growth and development in numerous research. Regulation of cell proliferation signaling by the Wnt/β-catenin and hedgehog pathways is significant in the etiology of several malignancies, including gastric cancer (Krishnamurthy & Kurzrock, 2018; Salaritabar et al., 2019). ZnRF3 protein, an E3 ubiquitin ligase, is altered or removed in cancer and inhibits Wnt signaling (Jiang et al., 2015). ZnRF3 overexpression increases apoptosis, and fewer cancer cells are proliferating than usual (Hao et al., 2012). Overexpression of ZnRF3 significantly reduced Lgr5 (Wnt signaling component) and Gli1 (Hedgehog signaling component). Therefore, ZnRF3 may negatively affect Wnt and Hedgehog proliferation pathways, thereby regulating cancer progression (de Lau et al., 2014; Qin et al., 2015).

Based on the previously identified DE-URGs, we identified 13 optimal variables related to TCGA-STAD survival: BLMH, CUL4A, HCFC1, IDE, NFE2L2, OGT, POLB, PSMD12, RNF144A, TGFBR1, THOP1, TRAF2, and VHL. A new predictive model has been developed for STAD. According to the KM survival curve, patients with STAD who were classified as greater risk had a lower outcome than those who were classified as low risk, indicating that the risk rating system can discriminate effectively between STAD patients’ prognoses. In addition, the model has a certain predictive accuracy.

Notably, RNF144A from the model was a potent oncogene like previously published ones (Wang et al. 2020). It has been discovered that the gene RNF144A is involved in cancer. Another name for it is ring finger protein 144A. This gene, which is found on chromosome 2q37.1, produces a protein that is an E3 ubiquitin ligase from the RBR (RING-between-RING) family. It has been discovered that RNF144A is involved in several biological functions, such as cell cycle regulation and protein degradation (Li et al., 2022). Regarding cancer, RNF144A has been connected to several cancer kinds, including colorectal, stomach, and breast cancers. Research has demonstrated that RNF144A expression is frequently dysregulated in cancer cells and that either overexpression or downregulation can significantly impact the development and spread of tumors (Ho & Lin, 2018; Li et al., 2021). Through ubiquitin ligase activity-dependent modulation of HSPA2′s stability and carcinogenic properties, RNF144A suppresses tumors in breast cancer (Yang et al., 2020). RNF144A inhibits the growth of tumors and the characteristics of ovarian cancer stem cells by controlling the degradation of LIN28B via the ubiquitin-proteasome pathway (Li et al., 2022). RNF144A targets YY1 for proteasomal degradation to decrease the production of GMFG, hence acting as a tumor suppressor in breast cancer (Zhang et al., 2022). RNF144A maintains EGFR signaling to encourage cancer cell proliferation that is dependent on EGF (Ho & Lin, 2018). In our study, RNF144A could predict the chemotherapy response of perhexiline, TAF1_5496_1732, trifluoperazine, and PF-00299804_363. Besides, RNF144A is proven to be a potential immunotherapy determinant. Lack of RNF144A increases bladder carcinogenesis caused by carcinogens and the stability of the PD-L1 protein (Ho et al., 2021). The process of tagging proteins with ubiquitin molecules to mark them for cellular functions such as breakdown is known as ubiquitination. Conversely, immunotherapy is a form of treatment that targets cancer or other disorders by stimulating the body’s immune system. There is mounting evidence that ubiquitination affects immunotherapy efficacy and is essential for controlling the immune response. For instance, immunotherapy medications target immunological checkpoint proteins like CTLA-4 and PD-1, whose ubiquitination regulates expression. The stability and function of immune cells, such as T cells and natural killer cells, which are important components of immunotherapy, can also be impacted by ubiquitination. It is reasonable that RNF144A could predict immunotherapy response. RNF144A could facilitate tumor progression and macrophage aggregation. It has been discovered that RNF144A specifically controls the polarization of macrophages toward an anti-inflammatory phenotype or M2 macrophages. To do this, RNF144A encourages the breakdown of a protein known as TRAF6, which is necessary for activating pro-inflammatory signaling pathways.

We established a more reliable prediction method for clinicians and provided a target, RNF144A, for individualized treatment of STAD. The specific mechanisms of immunotherapy prediction of RNF144A is expected to be further validated by more in vitro and in vivo experiments. Our model is likely to help with the clinical management of STAD patients. However, our sample data may need more comprehensive to reduce predictive bias. We will continue to study the role of other model genes and refine our prognostic model. We will continue researching to discover the underlying molecular mechanisms and validate our analytical results.

Supplemental Information

Supplemental Information 1 Supplementary Tables

Click here for additional data file.

Supplemental Information 2 Supplementary Figures

Click here for additional data file.

Supplemental Information 3 Raw Data for R codes and matrices of generating Figs. 1–7 and Figs. S1–S5

Click here for additional data file.

Supplemental Information 4 Raw Data for Fig. 8A

qPCR assay.

Click here for additional data file.

Supplemental Information 5 Raw Data for Fig. 8B

CCK-8 assay.

Click here for additional data file.

Supplemental Information 6 Raw Data for Figs. 8C and 8D

EdU assay.

Click here for additional data file.

Supplemental Information 7 Raw Data for Figs. 8E and 8F

Transwell assay.

Click here for additional data file.

Supplemental Information 8 Raw Data for Figs. 8G and 8H

Coculture Transwell assay.

Click here for additional data file.

Supplemental Information 9 MIQE checklist for RT-qPCR

Click here for additional data file.

Additional Information and Declarations

Competing Interests

Author Contributions

Data Availability

The authors declare there are no competing interests.

Shuai Shao conceived and designed the experiments, performed the experiments, analyzed the data, prepared figures and/or tables, authored or reviewed drafts of the article, and approved the final draft.

Yang Sun conceived and designed the experiments, performed the experiments, analyzed the data, prepared figures and/or tables, authored or reviewed drafts of the article, and approved the final draft.

Dongmei Zhao conceived and designed the experiments, performed the experiments, analyzed the data, prepared figures and/or tables, authored or reviewed drafts of the article, and approved the final draft.

Yu Tian conceived and designed the experiments, performed the experiments, analyzed the data, prepared figures and/or tables, authored or reviewed drafts of the article, and approved the final draft.

Yifan Yang conceived and designed the experiments, performed the experiments, analyzed the data, prepared figures and/or tables, authored or reviewed drafts of the article, and approved the final draft.

Nan Luo conceived and designed the experiments, performed the experiments, analyzed the data, prepared figures and/or tables, authored or reviewed drafts of the article, and approved the final draft.

The following information was supplied regarding data availability:

The raw measurements, R codes, matrices and the raw data of the in vitro validation are available in the Supplementary Files.

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
