# Peer review of "A ubiquitination-related risk model for predicting the prognosis and immunotherapy response of gastric adenocarcinoma patients"

_PeerJ, doi:10.7717/peerj.16868_

## Round 0.1 · original submission · Major Revisions

Please revise the manuscript as the reviewers suggested.

**Language Note:** The review process has identified that the English language must be improved. PeerJ can provide language editing services - please contact us at copyediting@peerj.com for pricing (be sure to provide your manuscript number and title). Alternatively, you should make your own arrangements to improve the language quality and provide details in your response letter. – PeerJ Staff

·

Basic reporting

1. The discussions pertaining to the results associated with RNF144A in the manuscript could be made more accessible by providing more detailed explanations. I suggest that the authors revise the manuscript to address this issue.

2. The figure legends accompanying Figures 5-7 would greatly benefit from more detailed explanations, specifically by providing additional illustrations that better depict the content of the panels. I suggest that the authors revise the manuscript to address this issue.

3. It would greatly benefit the manuscript if a fluent English-speaker were to polish the language used throughout. This would ensure a higher level of clarity and fluency in the text.

4. The authors could rearrange the figure panels in Figure 6 for better aesthetics. Besides, please enlarge the font size in Figure 5 and Figure 7.

Experimental design

1. I kindly request that you delve deeper into the mutation characteristics associated with the prognostic signature. It would greatly enhance the understanding and significance of the study.

2. The citation of R packages in the method section should be unified. For example, the R package limma should be cited in a consistent manner.

Validity of the findings

It has come to my attention that certain references cited in the study are outdated. To ensure the novelty and relevance of the research, I kindly suggest that the authors include more recent papers published after 2020 in their reference list to support their conclusion.

Additional comments

This study establishes a more reliable prediction method for clinicians and provides a target for individualized treatment of STAD. This study confirmed the oncogenic roles and immunotherapy prediction values of RNF144A. The manuscript could be considered for publication after following the improvements.

Reviewer 2 ·

Basic reporting

no comment

Experimental design

no comment

Validity of the findings

no comment

Additional comments

In this research, the authors have created a genetic model based on URGs that can appropriately gauge a patient's prognosis and immunotherapy response, providing clinicians with a reliable tool for prognostic assessment and supporting clinical treatment decisions. This is an interesting study. However, I have some suggestions regarding the current content.
1. RNF144A is a gene that has been found to play a significant role in pathogenic processes. Please introduce more about the pathogenic roles and molecular functions of RNF144A.
2. Ubiquitination, a post-translational modification process, has been increasingly recognized for its potential connections to immunotherapy. Please discuss the potential connections between ubiquitination and immunotherapy in this study.
3. Please add more sufficient explanations for performing some bioinformatics analyses. To ensure accurate and reliable results, it is essential to provide sufficient explanations for the chosen bioinformatics analyses.
4. Please give more details to the method section. In this way, researchers can enhance the reproducibility and transparency of their work.

·

Basic reporting

In this study, the authors created a model based on several prognostic genes and clinical indicators to predict OS in STAD patients. The TCGA dataset was screened to identify DE-URGs, and the 13 best prognostic gene groups were identified by Cox proportional hazards regression and LASSO univariate hazards regression analyses. Gene expression was multiplied by multivariate Cox coefficients to calculate risk scores. Perform internal/external validation to validate the risk scoring model. Nomograms were created by combining risk scores with clinical parameters and evaluated by calibration plots. DE-URGs were analyzed for their potential biological pathways using functional enrichment analysis. They further proved the oncogenic roles of the prognostic gene, RNF144A, using sufficient in vitro validation. Although the study is well-designed, it would still benefit from some improvements.
-Please add more necessary information to the method section of in vitro validation, such as reagents and procession concentration.
-The statistical analysis is missing in the method section.
-Besides immunotherapy prediction, please conduct the chemotherapy drug prediction of RNF144A.
-Please write more perspectives on this study.
-There are many typos and inappropriate expressions in the manuscript. Please check and correct these errors.

Experimental design

none

Validity of the findings

none

---

## Round 0.2 · Minor Revisions

The Section Editor pointed out that the authors need to do a better literature search in this area.

For example, ubiquitin in gastric cancer.
Zhongting Huang, Zhiyong Zhang, Yangyang Tu, Haibin He, Feng Qiu, Hailong Qian, Chunshu Pan. Integrated Single-Cell and Transcriptome Sequencing Analyses Develop a Ubiquitination-Associated Signature in Gastric Cancer and Identified OTULIN as a Novel Biomarker. Front. Biosci. (Landmark Ed) 2023, 28(11), 304. https://doi.org/10.31083/j.fbl2811304

Related to RNF144A: Yang, YL., Zhang, Y., Li, DD. et al. RNF144A functions as a tumor suppressor in breast cancer through ubiquitin ligase activity-dependent regulation of stability and oncogenic functions of HSPA2. Cell Death Differ 27, 1105–1118 (2020). https://doi.org/10.1038/s41418-019-0400-z

·

Basic reporting

no comment

Experimental design

no comment

Validity of the findings

no comment

Reviewer 2 ·

Basic reporting

no comment

Experimental design

no comment

Validity of the findings

no comment

Additional comments

no comment

·

Basic reporting

none

Experimental design

none

Validity of the findings

none

Additional comments

none

---

## Round 0.3 · Minor Revisions

I understand that you have provided the wrong files for Fig 8 and the associated data.

Please upload the correct files in your next revision.

---

## Round 0.4 · accepted · Accept

This manuscript can be accepted now.